

# Analysis of cerebral Interleukin-6 and tumor necrosis factor alpha patterns following different ventilation strategies during cardiac arrest in pigs

Miriam Renz[1], Lea Müller[1], Manuel Herbst[2], Julian Riedel[1], Katja Mohnke[1], Alexander Ziebart[1] and Robert Ruemmler[1]

[1] Department of Anesthesiology, Johannes-Gutenberg Universität Mainz, Mainz, Germany
[2] Institute for Medical Biometry, Epidemiology and Information Technology, University Medical Center of the Johannes Gutenberg Universität, Mainz, Germany

## ABSTRACT

Hypoxia-induced neuroinflammation after cardiac arrest has been shown to be mitigated by different ventilation methods. In this prospective randomized animal trial, 35 landrace pigs were randomly divided into four groups: intermittent positive pressure ventilation (IPPV), synchronized ventilation 20 mbar (SV 20 mbar), chest compression synchronized ventilation 40 mbar (CCSV 40 mbar) and a control group (Sham). After inducing ventricular fibrillation, basic life support (BLS) and advanced life support (ALS) were performed, followed by post-resuscitation monitoring. After 6 hours, the animals were euthanized, and direct postmortem brain tissue samples were taken from the hippocampus (HC) and cortex (Cor) for molecular biological investigation of cytokine mRNA levels of Interleukin-6 (IL-6) and tumor necrosis factor alpha (TNFα). The data analysis showed that CCSV 40 mbar displayed low TNFα mRNA-levels, especially in the HC, while the highest TNFα mRNA-levels were detected in SV 20 mbar. The results indicate that chest compression synchronized ventilation may have a potential positive impact on the cytokine expression levels post-resuscitation. Further studies are needed to derive potential therapeutic algorithms from these findings.

## INTRODUCTION

The clinical management of patients undergoing cardiac arrest poses a significant challenge in the hospital as well as in the emergency medical setting (*Gräsner et al., 2020*). Even after the initial critical phase is survived by patients and a return of spontaneous circulation (ROSC) can be achieved, hypoxia-induced organ dysfunctions and systemic inflammatory responses may still threaten patient outcomes (*Luh et al., 2011*; *Björklund et al., 2014*). Interleukin-6 (IL-6) and tumor necrosis factor alpha (TNFα) have been shown to play important roles as humoral factors in this context (*Kamuf et al., 2020*). Previous studies have demonstrated that elevated levels of these cytokines are associated with increased morbidity and mortality in patients who have undergone cardiopulmonary resuscitation

Corresponding author
Robert Ruemmler,
robert.ruemmler@email.de

(CPR) (*Högler et al., 2010*). IL-6 has been found to be involved in *e.g.*, tissue hypoxia after cardiac arrest and has been shown to be independently associated with poor outcomes in unconscious out-of-hospital cardiac-arrest patients (*Bro-Jeppesen et al., 2015*; *Tanaka & Kishimoto, 2018*; *Meyer et al., 2020*). Due to cardiac arrest and cerebral ischemia also the expression of TNFα gets up-regulated (*Niemann et al., 2013*; *Palmer et al., 2022*).

There is an extensive body of research regarding lung-protective ventilation in intensive care settings (*Brower et al., 2000*; *Brower et al., 2004*). Ventilation strategies during resuscitation efforts still are a highly researched topic and a lot of different strategies are being evaluated. However, it has been demonstrated, that low-tidal volume ventilation during CPR might attenuate cytokine release and mitigate neuroinflammation (*Ruemmler et al., 2020*; *Ruemmler et al., 2018*).

The purpose of this prospective randomized animal trial was to evaluate the mRNA-expression patterns of IL-6 and TNFα in pigs who underwent CPR with different ventilation strategies and to investigate their potential influence on the post-resuscitation systemic inflammatory response in order to potentially identify new treatment approaches.

## MATERIALS & METHODS

For this prospective randomized animal trial, after gaining approval of the State and Institutional Animal Care Committee Rhineland Palatine (approval no. G 20-1-065), 35 landrace pigs at age 12–16 weeks and with a weight of 29–34 kg were examined.

### Animal protocol

The general trial set-up up to the intervention was described in detail before (*Renz et al., 2022*). In short, the animals were anesthetized, intubated and instrumented with intravascular catheters. Prior to the intervention, the animals were randomized into three intervention groups: intermittent positive pressure ventilation (IPPV), synchronized ventilation 20 mbar (SV 20 mbar) and chest compression synchronized ventilation 40 mbar (CCSV 40 mbar) ($n = 10$ per group) or into a control group: Sham ($n = 5$), which did not receive CPR (Table 1). To start the intervention a fibrillation catheter (VascoMed, Binzen, Germany) was placed transvenously into the right atrium. Ventricular fibrillation was induced using a fibrillation frequency between 50 and 200 Hertz (Hz), followed by a 2-minute controlled no-flow and no ventilation period. Then, basic life support (BLS) was started with mechanical chest compressions by the LUCAS 2 system (Stryker, Kalamazoo, MI, USA) at a rate of 100 compressions/minute. The ventilation was carried out with a special ventilation device (type MEDUMAT Standard2, Weinmann Emergency Medical Technology GmbH + Co. KG, Hamburg, Germany). After 8 min of BLS, advanced life support (ALS) was performed using biphasic defibrillation with an energy of 200 Joule. After the first defibrillation, 1 mg epinephrine and 15 international units (IU) vasopressin were administered, and they were re-administered after each rhythm analysis every 2 min. After the third rhythm analysis, 150 mg amiodarone were injected. A maximum of seven rhythm analyses and six defibrillations were performed. Upon ROSC, ventilation was performed by an intensive care respirator (Engstroem care station, GE Healthcare, Munich, Germany) and ventilation was adjusted according to the ARDS network mechanical ventilation

**Table 1  Intervention groups and their parameters during resuscitation.**

| Groups | IPPV | SV 20 mbar | CCSV 40 mbar |
|---|---|---|---|
| Ventilation mode | Intermittent positive pressure ventilation | Experimental synchronized ventilation | Chest compression synchronized ventilation |
| Respiratory rate | 10 /min | 100 /min (mechanical chest compression frequency) | 100 /min (mechanical chest compression frequency) |
| I:E | 1:1 | 1:1 | 1:1 |
| Inspiratory time | 1.5 s | ~200 ms | ~200 ms |
| Peak pressure | 40 mbar | 20 mbar | 40 mbar |
| Positive endexspiratory pressure | 5 mbar | 3 mbar | 3 mbar |
| Fraction of inspired oxygen | 1.0 | 1.0 | 1.0 |
| Tidal volume | 10 ml/kgBW | ~2–3 ml/kgBW | ~4–5 ml/kgBW |
| Trigger | 5 | 5 | 5 |

protocol (*Brower et al., 2004*). The animals were kept under general anesthesia and were continuously monitored. Blood pressure was kept above 60 mmHg, using norepinephrine when necessary. Blood gas analyses were performed every hour post ROSC. After 6 h the trial was terminated by euthanizing the animals with a high dose of propofol (200 mg) and potassium chloride (40 mmol).

## RNA extraction and PCR

Directly postmortem brain tissue samples were taken from the hippocampus (HC) and cortex (Cor) for molecular biological investigations of cytokine mRNA-expression levels of proinflammatory IL-6 and TNFα. After removal, the tissue samples were snap frozen in liquid nitrogen and stored at −80 °C. For the quantitative determination of mRNA-levels of IL-6 and TNFα RNA extraction (RNeasy Plus Universal Mini Kit, Qiagen, Hilden, Germany) and *via* the intermediate step of cDNA creation (Quantitect Reverse Transcription, Qiagen GmbH, Hilden, Germany) real-time polymerase chain reaction (Absolute Blue qPCR SYBR green Mix AB-4166, Thermo Fisher Scientific, Waltham, MA, USA) was performed and cyclophilin A (peptidyl-prolyl isomerase A, PPIA) was used as a reference. Evaluations were executed using the Lightcycler 480 system (LightCycler, Roche, Mannheim, Germany). Measurements were carried out according to manufacturer's instructions. The applied primer sequences are summarized in Table 2.

## Statistical analysis

Statistical analyses were performed with the software R (*R Core Team, 2022*). Measurements of cytokine levels are summarized and reported as median (Q1–Q3) and visualized as box plots. We used rank based nonparametric statistical tests and $p$-values <0.05 are considered statistically significant. Kruskal-Wallis one-way ANOVA was applied to compare cytokine measurements among the four groups (IPPV, SV 20 mbar, CCSV 40 mbar and Sham). We performed Dunn's test for *post hoc* pairwise comparisons of ranked data und used the Bonferroni–Holm procedure to adjust $p$-values for multiplicity. Wilcoxon's signed-rank test for paired data was carried out to compare IL-6 mRNA- with TNFα mRNA-levels and

**Table 2** Real-time PCR primers.

| PCR assay | Oligonucleotide sequence | Version |
|---|---|---|
| sIL-6 | S: CCAATCTgggTTCAATCAggA | NM_214399 |
| | A: gTggTggCTTTgTCTggATTC | |
| TNF3 | S: CCCAgAAggAAgAgTTTCCA | NM_214022 |
| | A: CggCTTTgACATTggCTACA | |
| PPIA | S: CTTTCACAgAATAATTCCAggATT | NM_214353 |
| | A: ggACAAgATgCCAggACC | |

Notes.

S, sense primer; A, anti-sense primer; IL-6, Sus scrofa interleukin 6 (interferon, beta 2), mRNA; TNF α, Sus scrofa tumor necrosis factor (TNF superfamily, member 2), mRNA; PPIA, Sus scrofa peptidylprolyl isomerase A (cyclophilin A) (PPIA), mRNA.

to compare measurements from different regions (HC, Cor). Additionally, we computed Spearman's rank correlation coefficients and the corresponding p–values were obtained from an asymptotic approximation.

Data are pre-processed beforehand as follows: first, cytokine mRNA measurements were re-scaled by a factor of 1,000 to avoid reporting infinitesimally small numbers. Second, for our analysis, we included only those animals that reached ROSC and where all measurements could be observed.

# RESULTS

In total 35 landrace pigs were examined, of which five animals were in the sham group and did not receive intervention. Of the animals which were resuscitated, 22 animals reached ROSC. For statistical analysis 25 animals could be included: 21 animals which reached ROSC and four sham animals.

## Comparison of ventilation modes

Between the four groups some differences were seen. The comparison of the different ventilation modes showed that Kruskal–Wallis one-way ANOVA yields a statistically significant difference ($p = 0.01$ KW, Fig. 1) for TNF$\alpha$ mRNA-levels measured in HC. Here, the CCSV 40 mbar group showed by far the lowest TNF$\alpha$ mRNA-levels and the narrowest IQR (Table 3, Fig. 1). This explains the two significant ($p < 0.05$) *post hoc* comparisons of CCSV 40 mbar with SV 20 mbar and the sham group. For the SV 20 mbar group, we observed the highest TNF$\alpha$ mRNA-levels. These high levels could be detected in the cortex as well as in HC.

## Comparison of cytokines

Applying Wilcoxon's signed rank test, we observed that IL-6 mRNA-levels were significantly higher than TNF$\alpha$ mRNA-levels in the HC as well as in the Cor ($p < 0.0001$, Fig. 2). The correlation of IL-6 mRNA- and TNF$\alpha$ mRNA-levels was stronger in the Cor (total: $r = 0.51$, $p = 0.01$, Table 4) than in the HC (total: $r = 0.2$, $p = 0.34$, Table 4). In the cortex this trend could be seen primarily in the SV 20 mbar group (Table 4).

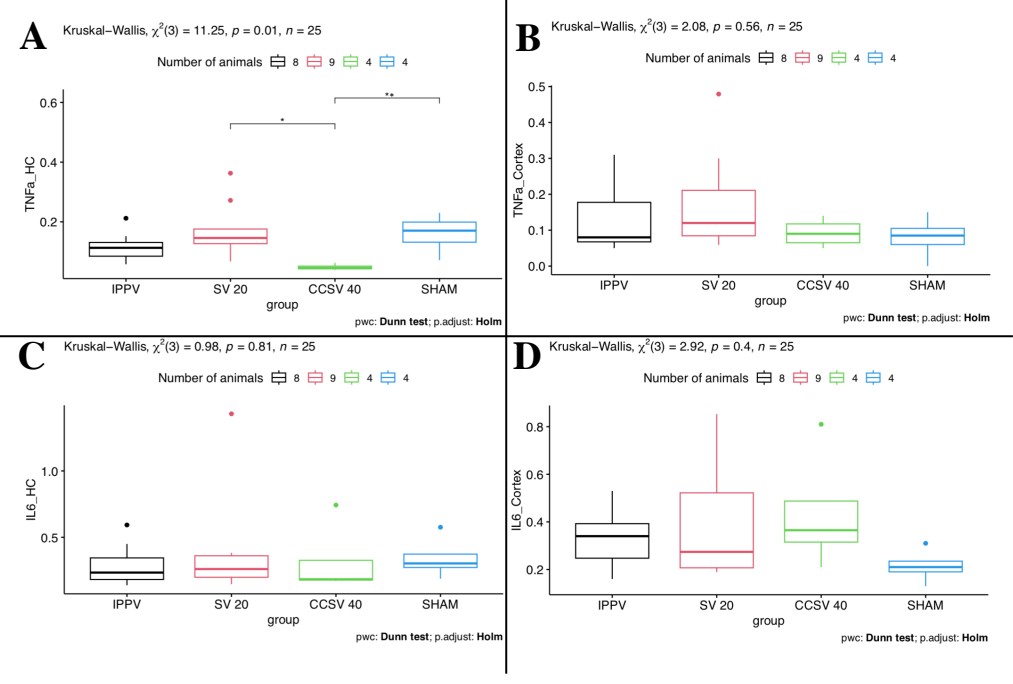

**Figure 1  Kruskal–Wallis one-way ANOVA (comparison of the four groups).** Data presented as box-plots. IPPV (intermittent positive pressure ventilation), SV 20 (synchronized ventilation 20 mbar), CCSV 40 (chest compression synchronized ventilation 40 mbar), Sham (control group), TNFα (tumor necrosis factor alpha), IL-6 (Interleukin-6), HC (Hippocampus). A: Kruskal–Wallis (KW) showed significant difference ($p = 0.01$) for TNFα mRNA-levels in HC. An asterisk (*) indicates a significant ($p < 0.05$) *post hoc* comparison of CCSV 40 mbar *vs.* SV 20 mbar; two asterisks (**) indicate significant ($p < 0.05$) *post hoc* comparison of CCSV 40 mbar *vs.* sham. B–D: No significant differences were seen.

**Table 3  Descriptive summary statistics.**

| Variables | IPPV (N = 8) | SV 20 (N = 9) | CCSV 40 (N = 4) | SHAM (N = 4) | Total (N = 25) |
|---|---|---|---|---|---|
| **TNFa_HC** | | | | | |
| mean ± sd | 0.12 ± 0.049 | 0.17 ± 0.092 | 0.048 ± 0.01 | 0.16 ± 0.067 | 0.13 ± 0.078 |
| median (Q1, Q3) | 0.11 (0.081, 0.14) | 0.15 (0.13, 0.18) | 0.046 (0.041, 0.055) | 0.17 (0.11, 0.21) | 0.12 (0.072, 0.17) |
| **IL6_HC** | | | | | |
| mean ± sd | 0.29 ± 0.16 | 0.38 ± 0.4 | 0.32 ± 0.28 | 0.34 ± 0.16 | 0.34 ± 0.28 |
| median (Q1, Q3) | 0.23 (0.18, 0.38) | 0.26 (0.2, 0.36) | 0.18 (0.18, 0.47) | 0.3 (0.24, 0.44) | 0.24 (0.19, 0.36) |
| **TNFa_Cortex** | | | | | |
| mean ± sd | 0.13 ± 0.092 | 0.17 ± 0.14 | 0.092 ± 0.04 | 0.08 ± 0.062 | 0.13 ± 0.1 |
| median (Q1, Q3) | 0.08 (0.065, 0.18) | 0.12 (0.084, 0.21) | 0.09 (0.06, 0.12) | 0.085 (0.04, 0.12) | 0.09 (0.07, 0.15) |
| **IL6_Cortex** | | | | | |
| mean ± sd | 0.33 ± 0.12 | 0.4 ± 0.24 | 0.44 ± 0.26 | 0.21 ± 0.074 | 0.35 ± 0.19 |
| median (Q1, Q3) | 0.34 (0.23, 0.4) | 0.27 (0.21, 0.52) | 0.36 (0.28, 0.59) | 0.21 (0.17, 0.26) | 0.29 (0.21, 0.39) |

**Notes.**

IPPV, intermittent positive pressure ventilation; SV 20, synchronized ventilation 20 mbar; CCSV 40, chest compression synchronized ventilation 40 mbar; Sham, control group; TNFα, Tumor necrosis factor alpha; IL-6, Interleukin-6; Cor, Cortex; HC, Hippocampus; sd, standard deviation.

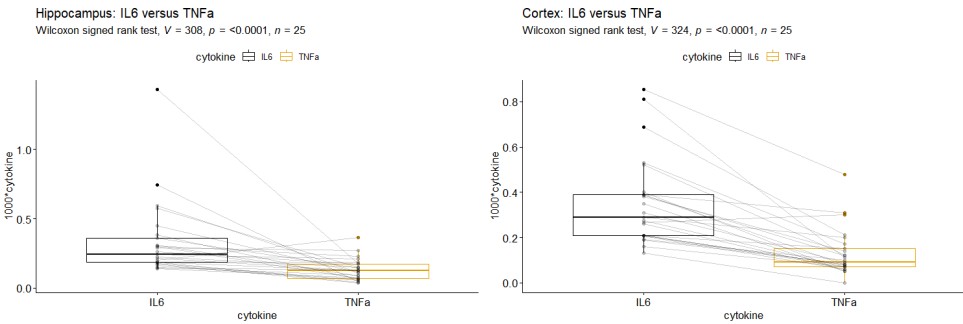

**Figure 2  Comparison of tumor necrosis factor alpha mRNA *vs.* Interleukin-6 mRNA in (A) Hippocampus and (B) Cortex.** Interleukin-6 (IL-6), tumor necrosis factor alpha (TNFα). A + B: IL-6 mRNA-levels were significantly higher than TNFα mRNA-levels ($p < 0.0001$).

**Table 4  Tumor necrosis factor alpha—Interleukin-6: Spearman correlation coefficients in Cortex *vs.* Hippocampus.**

| region | IPPV ($n = 8$) | SV 20 ($n = 9$) | CCSV 40 ($n = 4$) | SHAM ($n = 4$) | Total ($n = 25$) |
|---|---|---|---|---|---|
| Cortex | 0.19 ($p = 0.66$) | 0.67 ($p = 0.06$) | 0.6 ($p = 0.42$) | 0.63 ($p = 0.37$) | 0.51 ($p = 0.01$) |
| Hippocampus | 0.32 ($p = 0.43$) | 0.1 ($p = 0.81$) | 0.4 ($p = 0.75$) | 0 ($p = 1$) | 0.2 ($p = 0.34$) |

**Notes.**

IPPV, intermittent positive pressure ventilation; SV 20, synchronized ventilation 20 mbar; CCSV 40, chest compression synchronized ventilation 40 mbar; Sham, control group.

Significant correlation of total cytokine mRNA-levels were seen in the cortex ($p = 0.01$). SV 20 mbar showed also a significant correlation in the cortex ($p = 0.06$).

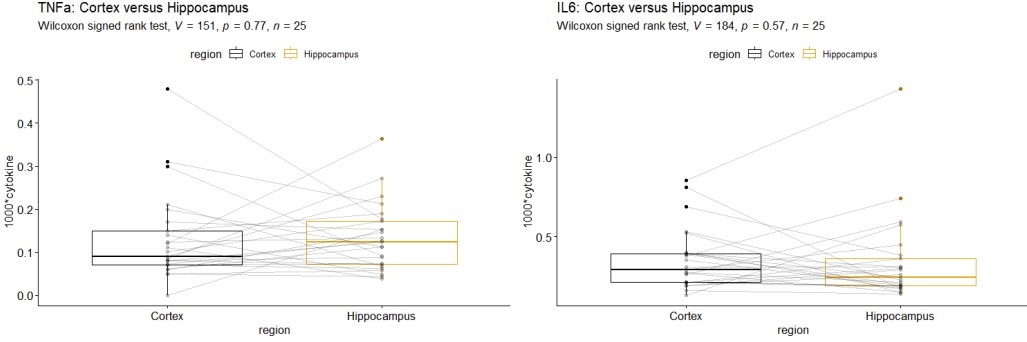

**Figure 3  Comparison of Hippocampus *vs.* Cortex for (A) Interleukin-6 mRNA and (B) tumor necrosis factor alpha mRNA.** Interleukin-6 (IL-6), tumor necrosis factor alpha (TNFα). A +B: No significant differences of cytokine mRNA-levels were seen in the two brain regions.

## Comparison of brain regions

According to Wilcoxon's signed rank test, there was no significant difference between cytokine mRNA-levels from the cortex and those from the HC, neither for TNFα nor for IL-6 (Fig. 3). We found a remarkably high correlation TNFα mRNA-levels in the HC and cortex in the IPPV group ($r = 0.77$, $p = 0.03$, Table 5) which might be due to multiplicity.

**Table 5  Hippocampus—Cortex: Spearman correlation coefficients for Interleukin-6 *vs.* tumor necrosis factor alpha.**

| cytokine | IPPV ($n = 8$) | SV 20 ($n = 9$) | CCSV 40 ($n = 4$) | SHAM ($n = 4$) | Total ($n = 25$) |
|---|---|---|---|---|---|
| IL6 | 0.08 ($p = 0.84$) | 0.17 ($p = 0.68$) | −0.2 ($p = 0.92$) | −0.32 ($p = 0.68$) | 0.02 ($p = 0.93$) |
| TNFa | 0.77 ($p = 0.03$) | −0.18 ($p = 0.64$) | 0 ($p = 1$) | 0.6 ($p = 0.42$) | 0.29 ($p = 0.16$) |

Notes.

IPPV, intermittent positive pressure ventilation; SV 20, synchronized ventilation 20 mbar; CCSV 40, chest compression synchronized ventilation 40 mbar; Sham, control group.
Strong correlation of TNFα mRNA-levels in the Hippocampus and Cortex in the IPPV group ($p = 0.03$).

## DISCUSSION

This study contributes to the limited understanding of neuroinflammatory mechanisms and cytokine expression patterns in pigs, which may be relevant for clinical outcomes following hypoxia and hypoperfusion caused by cardiac arrest. In our prospective randomized large animal model, we observed overall higher IL-6 mRNA-levels than TNFα mRNA-levels. Furthermore, our findings suggest that sophisticated ventilation strategies may have mitigating influences on the expression levels and release of these cytokines, because CCSV 40 mbar showed by far the lowest TNFα mRNA-levels and the narrowest IQR in the HC.

Cerebral tissue inflammation has been shown to play a significant role in the pathophysiology of cardiac arrest and post-cardiac arrest brain injury (PCABI) (*Sandroni, Cronberg & Sekhon, 2021*). Recent scientific findings suggest that activation of the immune system and release of pro-inflammatory cytokines after cardiac arrest can lead to secondary brain injury, exacerbating the initial injury caused by ischemia and reperfusion (*Sandroni, Cronberg & Sekhon, 2021*; *Björklund et al., 2014*; *Rocha-Ferreira et al., 2017*; *Graber, Costine & Hickey, 2015*). IL-6 and TNFα expression have been found to be involved in inflammation due to tissue hypoxia after cardiac arrest (*Bro-Jeppesen et al., 2015*; *Tanaka & Kishimoto, 2018*; *Meyer et al., 2020*; *Niemann et al., 2013*; *Palmer et al., 2022*). Apart from numerous studies conducted by our own research group, there have been relatively few investigations examining the specific distribution of proinflammatory cytokines in different brain regions in the porcine model following resuscitation (*Ruemmler et al., 2020*). Previous data indicate that in humans who did not survive cardiac arrest, the HC exhibited the most severe neuronal damage as a result of hypoxia (*Björklund et al., 2014*). However, cerebral tissue vulnerable to ischemia was also identified in the cortex, cerebellum, and thalamus (*Nolan et al., 2008*; *Sandroni, Cronberg & Sekhon, 2021*; *Sekhon, Ainslie & Griesdale, 2017*). Additionally, data on hypoxic brain injury following cardiac arrest in pigs demonstrated that neuronal changes can be observed in both the Cor and the HC (*Högler et al., 2010*). It is important to note that the aforementioned studies often focused on neuronal damage following hypoxia, rather than specifically examining neuroinflammation. Further research is required to specifically evaluate neuronal inflammatory damage in pigs following hypoxia due to resuscitation. The results of our study should be considered exploratory, as more animals with return of spontaneous circulation (ROSC) would be needed to achieve statistical significance.

Elevated cytokine levels are found in different causes of ischemia. In ischemia caused by stroke increased IL-6 levels could be seen and data showed subsequently aggravated

inflammation and histopathologic findings (*Armstead et al., 2019*). Aside from these findings, Interleukin-1 was identified as a primary driver of inflammation and has been a focus especially of autoimmune disease research trials (*Konsman, 2022*). In PCABI, due to the ischemia-reperfusion injury, microglia get activated and secrete pro-inflammatory cytokines (such as IL-6 and interleukin 1-beta), which, *via* further intermediate steps, leads to injury to the cells of the neurovascular unit (*Sandroni, Cronberg & Sekhon, 2021*). This observation is consistent with our findings, showing that IL-6 mRNA-levels were overall significantly higher than TNFα mRNA-levels. However, it must be kept in mind that while significantly higher levels of IL-6 have been found, concentrations of TNFα in the picogram range can have powerful effects (*Palmer et al., 2022*; *Niemann et al., 2013*). Further studies will have to show, if our TNFα mRNA-levels, although they are significantly lower than the IL-6 mRNA-levels, have an impact on PCABI in swine.

The treatment of neuroinflammation remains a challenge. General treatment strategies aimed at mitigating neuroinflammation and modulating the immune response after cardiac arrest have shown promise in preclinical studies. These approaches include the administration of anti-inflammatory drugs, such as corticosteroids and anti-cytokine therapies (*Liu & Quan, 2018*), as well as modulation of the gut microbiome to reduce systemic inflammation (*Trichka & Zou, 2021*). Yet again, none of those approaches have been tested clinically after cardiac arrest and it remains unclear, if they are valid therapeutic strategies in these settings.

In our trial, we tested three different ventilatory interventions during cardiac arrest, further examining the influence of more nuanced inspiratory pressure settings on brain physiology and overall outcome post-ROSC. This continues a series of original research projects specifically targeting sophisticated novel ventilation modes and their effects on tissue inflammation, cardiac output and lung physiology (*Renz et al., 2022*; *Ruemmler et al., 2018*; *Ruemmler et al., 2021*; *Ruemmler et al., 2020*). The chest-compression synchronized ventilation mode in particular showed promising results in previous studies regarding oxygenation and potential end-organ perfusion improvement (*Kill et al., 2015*; *Kill et al., 2009*). We decided to combine these findings with our own results on the beneficial effects of particularly low-tidal volume ventilation and developed a novel mode with synchronized ventilation strokes using very low airway pressures. The initial results from a previous pilot trial (data presently under review) indicate no significant differences in terms of oxygenation and decarboxylation. In this trial, a lower ROSC-rate was observed in CCSV 40 mbar animals. This could suggest potential issues with higher inspiratory pressures due to overdistension or even fatal lung damage (*Fichtner et al., 2019*; *Cheifetz et al., 1998*; *Pinsky et al., 1983*). However, these observations do not align with the original evaluations by *Kill et al. (2015)* where no increase in lung damage was observed. Additionally, blood gas analyses during resuscitation hinted at better oxygenation under CCSV 40 and impaired decarboxylation during SV 20. While this could explain some of the interleukin expression differences, those data were inconsistent and showed large individual variance (see supplement), as is often seen during resuscitation experiments. In the presented trial, we detected the highest TNFα mRNA-levels in SV 20 mbar. CCSV 40 mbar showed low TNFα mRNA-levels, especially in the HC. While improved oxygen

supply might be a reason for this, further studies would be necessary with a focus on this context.

This study has some inherent limitations. First, investigating resuscitation and post-ROSC pathophysiology in a prospective manner is predominantly conducted through large animal trials. This often involves porcine trials, due to their anatomical similarities and organ sizes compared to humans, such as their large and gyrencephalic brains, which enable the identification of (sub-)cortical structures (*Cherry et al., 2015*; *Lind et al., 2007*; *Swindle et al., 2012*). However, no concise evidence has been published stating that cytokine patterns observed in pigs actually bear any resemblance or clinical relevance for human treatment.

Second, this trial, though being prospectively-randomized, was not statistically powered beforehand. This was due to the very limited data available from which to infer any expected effects, resulting in an exploratory trial design with corresponding data analyses. The identified correlations largely do not reach statistical significance and suggest that substantially larger numbers of animals might be necessary to validate these results. This poses logistical as well as financial challenges, which would be a reason to optimize future study designs to either yield more samples per animal or test for more cytokines simultaneously. For this trial, due to local restrictions, only two cytokines could be analyzed.

Third, resuscitation trials as a whole are difficult to conduct and often yield greatly varying results due to highly variable ROSC rates. Depending on the research model, these can range from 10–100% (*Cherry et al., 2015*; *Hüpfl, Selig & Nagele, 2010*). In this trial, more than twice as many animals survived in the SV 20 mbar group compared to the CCSV 40 mbar group, making direct comparisons of post-ROSC developments less viable. Again, larger animal numbers could counteract this problem. Additionally, our laboratory was not permitted to house the animals for an extended period of time, excluding the option to facilitate emergence after ROSC and precluding any functional analyses on awake animals after the trial.

However, the presented results still provide further insights into the generation and expression patterns of proinflammatory cytokines. Since the available evidence, especially during and after resuscitation, is extremely limited, these exploratory results can help to design further trials and potentially identify viable targets for post-ROSC treatment.

## CONCLUSIONS

In this trial of different ventilation modes during resuscitation in swine, we observed that IL-6 mRNA-expression was increased in the hippocampal and cortex regions than TNFα mRNA-expression. Additionally, we detected lower TNFα mRNA-expression in the CCSV 40 mbar group, while the highest TNFα mRNA-expression was seen in the SV 20 mbar group. This could indicate that CCSV 40 may have a potential positive impact on cytokine expression levels post-resuscitation.

Although the trial has some methodological limitations, the results contribute to a very limited body of knowledge regarding post-resuscitation pathophysiology and offers further insights into the generation and expression patterns of proinflammatory cytokines. These

exploratory results can help design further trials and potentially identify viable targets for novel post-ROSC treatment approches.

## ACKNOWLEDGEMENTS

Parts of this trial will be contained in the doctoral thesis of Lea Müller. We want to thank Dagmar Dirvonskis for her professional support.

### Funding

This work was supported by a personal grant of the German Research foundation (DFG grant no RU 2371/1) to Robert Ruemmler and an intramural grant to Miriam Renz. The funders had no role in study design, data collection and analysis, decision to publish, or preparation of the manuscript.

### Grant Disclosures

The following grant information was disclosed by the authors:
German Research foundation: RU 2371/1.

### Competing Interests

The synchronized ventilation devices were provided by Weinmann Medical unconditionally and for research purposes only. The authors declare there are no competing interests.

### Author Contributions

- Miriam Renz conceived and designed the experiments, performed the experiments, analyzed the data, prepared figures and/or tables, authored or reviewed drafts of the article, and approved the final draft.
- Lea Müller performed the experiments, analyzed the data, prepared figures and/or tables, and approved the final draft.
- Manuel Herbst analyzed the data, authored or reviewed drafts of the article, and approved the final draft.
- Julian Riedel performed the experiments, prepared figures and/or tables, and approved the final draft.
- Katja Mohnke performed the experiments, authored or reviewed drafts of the article, and approved the final draft.
- Alexander Ziebart conceived and designed the experiments, authored or reviewed drafts of the article, and approved the final draft.
- Robert Ruemmler conceived and designed the experiments, prepared figures and/or tables, authored or reviewed drafts of the article, and approved the final draft.

### Animal Ethics

The following information was supplied relating to ethical approvals (*i.e.*, approving body and any reference numbers):

The State and Institutional Animal Care Committee Rhineland Palatine approved the study (approval no. G 20-1-065).

## Data Availability

The raw data are available in the Supplemental Files.

## Supplemental Information

Supplemental information for this article can be found online at http://dx.doi.org/10.7717/peerj.16062#supplemental-information.

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
