# Peer review of "Analysis of cerebral Interleukin-6 and tumor necrosis factor alpha patterns following different ventilation strategies during cardiac arrest in pigs"

_PeerJ, doi:10.7717/peerj.16062_

## Round 0.1 · original submission · Major Revisions

Thanks for forwarding your manuscript to PeerJ Cardiovascular and Metabolic Disorders. After analyzing the reviews that make up one minor and two major impressions, I reject the manuscript for immediate publication. Several aspects mentioned by the reviewers require profound corrections, which, at the moment, make it unfeasible to accept. However, if the authors are interested, all observations/corrections must be strongly considered for the manuscript to be sent again.

Best Regards,
Sincerely

Reviewer 1 ·

Basic reporting

The authors report experimental data about the influence of different ventilation modes during resuscitation on neuroinflammation of the brain in a pig model.
These data are of great interest for further research and development in resuscitation science. The methodology is adequate for answering the question of the study and both statistical analysis as well as data presentation and conclusions are well done.
Nevertheless I would like to suggest some changes to clarify and further improve the manuscript.

Experimental design

Methods:
Line 70ff: Please describe the ventilation modes more detailed, what does SV exactly mean? Which parameters exactly were chosen (IPPV: Vt, Pmax; RR, e.g.) ?
Why was CCSV limited to 40mbar (as there are only studies and recommendations between 45 and 60mbar)?
Could you provide (measured) ventilator data or even blood gas analysis as well as perfusion (blood pressure) data?
Did the sham group get “no intervention (no mechanical CPR)” or only “no ventilation”? Please clarify.

Validity of the findings

Discussion:
Line 200ff: You worte:
In this trial, a lower ROSC-rate was observed in CCSV 40 mbar animals. This could suggest potential issues with higher inspiratory pressures due to overdistension or even fatal lung damage (Fichtner et al., 2019, Cheifetz et al., 1998, Pinsky et al.203 1983).

I do not agree with this conclusion, as the cited study about CCSV (PLoS One. 2015 May26;10(5):e0127759. doi: 10.1371/journal.pone.0127759) showed that even a peak pressure of 45mbar may be not sufficient for ventilation, whereas there is no proof that higher peak pressures with low tidal volumes during chest compressions are associated with adverse effects to lung tissue:

http://dx.doi.org/10.1016/j.resuscitation.2013.08.052

So I would conclude instead, that the lower ROSC rate in CCSV may be caused by to low peak inspiratory pressures, not by high pressures.

Please discuss this topic in your group based on existing data.

Reviewer 2 ·

Basic reporting

Analysis of cerebral Interleukin-6 and tumor necrosis factor alpha patterns following different ventilation strategies during cardiac arrest in pigs (#84595)

This paper is about prospective randomized animal trial, with landrace pigs that were randomly divided into 4 groups: Intermittent positive pressure ventilation (IPPV), synchronized ventilation 20 mbar (SV 20 mbar), chest compression synchronized ventilation 40 mbar (CCSV 40 mbar) and a control group (Sham). The animals were euthanized, and direct postmortem brain tissue samples were taken from the hippocampus (HC) and córtex (Cor) for molecular biological investigation of cytokine expression mRNA levels of Interleukin-6 (mRNA IL-6) and mRNA Tumor necrosis factor alpha (mRNATNF³). The data analysis showed that CCSV 40 mbar displayed low mRNATNF³ levels, especially in the HC, while the highest mRNA TNF³ levels were detected in SV 20 mbar. The results indicate that chest compression synchronized ventilation may have a potential positive impact on the cytokine expression levels postresuscitation.

This paper has great potential and brings intresting results properly reconigzed by the authors as exploratory. To make it more robust we suggest that the authors consider some sugestions.

Experimental design

Introdution section:
1. 46-48 lines - Search for pathology and immunology concepts theta explain the role of Il-6 and TNF in neuroinflamation. The mechanism for this injury exists and is explained in these sciences.
2. 48 line – use more recente references on the inflammatory induced by IL-6 and TNF.
3. line 50 – Brower et al., 2000 is not listed. Check that all cited references are listed.
4. 50 line – “…only scarce data is available on favorable ventilation strategies during resuscitation efforts…” There is a considerable bibliography, updating this positioning.
5. 53 line – The authors could enrich the paper with more recente references based on studies with COVID-19.

Materials & Methods section:
1. 98 line – “For the quantitative determination of IL-6 and TNF³…” Did the authors quantify the levels of cytokines IL-6 and TNF in brain tissue? Or just mRNA levels? If the cythokine level was measured, what thechnique was used? This paper only describes mRNA quantification.
2. 108 line – “Measurements of cytokine levels are summarized and reported as median..” Correct if only mRNA was dosed.
Measurements of mRNA cytokine levels….
3. 114 line – “IL-6 with TNF³ levels…” change to … mRNA IL6 with mRNA TNF³ levels…
4. 117 line - “cytokine measurements” changed to … mRNA cytokyne….
5. 105 line – “The applied primer sequences are summarized in Table 1” Were the primers constructed by the authors or were they selected based on the literature? If constructed, we suggest describing, otherwise refer.

Results section

In all section substitute IL-6 and TNF levels to IL-6 mRNA and TNFmRNA levels.

Discussion

We sugest to the authors consider that although significantly higher amounts of IL-6 were found, nanogram concentrations of TNF can exert impactful effects. Seek for discussion the potential of these two citokines whem compared concentrations.

Conclusions

In all section substitute IL-6 and TNF levels to IL-6 mRNA and TNFmRNA levels.


Figures:
In all section substitute IL-6 and TNF levels to IL-6 mRNA and TNFmRNA levels, included in chart legend.

Tables

Table 1.

Sense and anti-sense are the same as Forward and reverse. Unify namig in legend and table.
In all section substitute IL-6 and TNF levels to IL-6 mRNA and TNFmRNA levels.
Please, check table formatting.

Validity of the findings

no comment

Additional comments

Search for pathology and immunology concepts theta explain the role of Il-6 and TNF in neuroinflamation. The mechanism for this injury exists and is explained in these sciences.
Use more recente references on the inflammatory induced by IL-6 and TNF. In COVID-19 experience, many articles were produced.
Some errors in the M&M must be corrected throughout the article.

Reviewer 3 ·

Basic reporting

no comment

Experimental design

The methods not are described with sufficient details and information to replicate.
The methods described are not sufficient to meet the proposed objectives (L54-57). The autors does not describe some detalis minimum. The study design it is incomplete, its not describe for example; how the sample size was decided and the experimentals procedures is not described clearly, and mainly are not describe clearly all outcomes measures assessed.

Validity of the findings

The results have been provided not are robust, and statistically sound and controlled, and the conclusions are not clearly described. In the results the authors do not use as references studies describing hypoxia in production pigs.

---

## Round 0.2 · Major Revisions

Dear authors,

We are submitting the second round of manuscript reviews — several suggestions made by reviewers 1 and 3 needed to be clarified or considered in the new draft. Therefore, please, consider capital adopting the observations and suggestions again sent by the reviewers. Pay attention to the following aspects:

1) detail the ventilation mode of animals during the experimental procedures;

2) clarify how the clinical evaluations of post-resuscitation were conducted in animals;

3) provide the blood gas analysis data;

4) clarify the reason for including more than two experimental groups,

5) discuss the theory of inspiratory pressure (CCSV) and clarify the statements about Pmax 40mbar and insufficient ventilation according to the low rate of ROSC,

5) rewrite the conclusion according to the hypothesis and relevant findings.

Sincerely

Reviewer 1 ·

Basic reporting

none

Experimental design

1. An additional table describing the different ventilation modes was added. Nevertheless this table does not describe the ventilation mode presets in detail. Please report all presets (eg tidal volume in IPPV, trigger presets in SV/CCSV, Tinsp in all)

2. Please provide data of blod gas analyses. You wrote:
Blood gas analyses and pressure data did not show any differences between the two groups.
I do not understand, because you report about more than two groups.

Validity of the findings

Please discuss your theory aboout inspiratory pressure in CCSV as recomended before concerning potential lung damage as recommended before in your manuscript with respect to:
http://dx.doi.org/10.1016/j.resuscitation.2013.08.052
I do not agree with your theory, because Pmax of 40mbar may also lead to insufficient ventilation at all in CCSV expalining the lower ROSC rate.

Reviewer 2 ·

Basic reporting

The suggested changes were accepted and carried out meet the expectations for the publication of the article.

Experimental design

The suggested changes were accepted and carried out meet the expectations for the publication of the article.

Validity of the findings

The suggested changes were accepted and carried out meet the expectations for the publication of the article.

Additional comments

The suggested changes were accepted and carried out meet the expectations for the publication of the article.

Reviewer 3 ·

Basic reporting

The submission does not describe the results adequate to answer the hypothesis. Because it does not describe how the clinical evaluations of the pos pigs post-resuscitation were made and does not describe also the identify and aplication new treatment approaches.

Experimental design

The submission not clearly define the research question and the methods describ does not sufficient information to be reproducible by another investigator. The methods described the autores not sufficient to anwer the purpose. The methods does not clearly described, for example, the autors not described the clinical exames used the pigs our a news treatment approaches.

Validity of the findings

The data on which the conclusions are based does not are robust and statistically sound or controlled. The results described in the article not sufficient for drawing up an appropriate conclusion.

---

## Round 0.3 · Minor Revisions

Dear Dr. Ruemmler,

Consider the minor revisions according to Review 1. Please address these changes and resubmit.

Best regards

Reviewer 1 ·

Basic reporting

see rev 1

Experimental design

see rev 1

Validity of the findings

see rev1

Additional comments

The authors meanwhile adressed proposed changes and provided additional data. Unfortunately there are several obvious mistakes that needed to be corrected:
In the supplemental data sheet with blood gas analyses somtimes vent modes are (correctly?)described as
paCO2
IPPV
SV 20mbar
CCSV 40mbar
below and in the graph as
paO2 IPPV
CCSV 20
CCSV 40
CCSV 20 was nowhere described in the text. Please crarify.

response letter and table 1: Still missing in- and expiratory times: If CCSV is provided with the described ventilator, Tinsp ist independent of resp rate fixed to 205ms. Please also describe SV more detailed including exp. trigger.

---

## Round 0.4 · accepted · Accept

Dear authors,

After reviewing your manuscript based on your latest corrections, my decision is in favor of publication. Thank you for submitting your manuscript to PeerJ.